# Mechanical and Microstructural Anisotropy of Laser Powder Bed Fusion 316L Stainless Steel

**DOI:** 10.3390/ma15020551

**Published:** 2022-01-12

**Authors:** Zdeněk Pitrmuc, Jan Šimota, Libor Beránek, Petr Mikeš, Vladislav Andronov, Jiří Sommer, František Holešovský

**Affiliations:** Department of Machining, Process Planning and Metrology, Center of Advanced Aerospace Technology, Faculty of Mechanical Engineering, The Czech Technical University in Prague, 160 00 Prague, Czech Republic; jan.simota@fs.cvut.cz (J.Š.); libor.beranek@fs.cvut.cz (L.B.); p.mikes@fs.cvut.cz (P.M.); vladyslav.andronov@fs.cvut.cz (V.A.); jiri.sommer@fs.cvut.cz (J.S.); frantisek.holesovsky@fs.cvut.cz (F.H.)

**Keywords:** additive manufacturing, laser powder bed fusion, AISI 316L, anisotropy, porosity, solution annealing, melting level, focus level parameter

## Abstract

This paper aims at an in-depth and comprehensive analysis of mechanical and microstructural properties of AISI 316L austenitic stainless steel (W. Nr. 1.4404, CL20ES) produced by laser powder bed fusion (LPBF) additive manufacturing (AM) technology. The experiment in its first part includes an extensive study of the anisotropy of mechanical and microstructural properties in relation to the built orientation and the direction of loading, which showed significant differences in tensile properties among samples. The second part of the experiment is devoted to the influence of the process parameter focus level (FL) on mechanical properties, where a 48% increase in notched toughness was recorded when the level of laser focus was identical to the level of melting. The FL parameter is not normally considered a process parameter; however, it can be intentionally changed in the service settings of the machine or by incorrect machine repair and maintenance. Evaluation of mechanical and microstructural properties was performed using the tensile test, Charpy impact test, Brinell hardness measurement, microhardness matrix measurement, porosity analysis, scanning electron microscopy (SEM), and optical microscopy. Across the whole spectrum of samples, performed analysis confirmed the high quality of LPBF additive manufactured material, which can be compared with conventionally produced material. A very low level of porosity in the range of 0.036 to 0.103% was found. Microstructural investigation of solution annealed (1070 °C) tensile test samples showed an outstanding tendency to recrystallization, grain polygonization, annealing twins formation, and even distribution of carbides in solid solution.

## 1. Introduction

Additive manufacturing (AM) has experienced unprecedented development in recent years, and it has gained a regular role among production technologies. AM allows the application of completely different, very complex, and non-traditional CAD approaches to the design of parts that are otherwise very limited by conventional manufacturing technologies. The growing availability of additive systems really opens the door to modern design and manufacturing tools, such as rapid prototyping, rapid manufacturing, or rapid tooling [1]. Functional prototypes can now be produced within a short time with less effort directly from CAD design data, leading to reduced development costs and a lower environmental burden [2].

With laser powder bed fusion (LPBF) technology, units of material feedstock in the form of powder are distributed layer by layer in a powder bed and fused in desired regions together layer by layer using the thermal energy of a scanning laser beam. Layers of powder are applied to the desired thickness by a recoater. LPBF is single step AM intended for processing, with rare exceptions, of single material [3,4,5,6]. LPBF triggers an extraordinary versatility to geometry and material design, being considered a near-net-shape technology as well as an exceptional technique to create functionally graded materials [7] and complex components with individualized local functional requirements [8,9]. Over the last decade, sintering parameters have been optimized, and porosity has been significantly eliminated. This has made it possible to produce almost fully dense components with mechanical properties comparable to conventionally produced materials [4,5,10,11] resulting in several medical, mold making, and aerospace and aviation application. A particularly successful mastery of the printing of titanium and cobalt chromium alloys has enabled the production of dental, structural, joint, and cranial implants, without which some reconstructive traumatology surgeries would not be possible at all [12,13,14,15,16].

Despite all the advances in additive technologies and the growing number of applications, many attempts to qualify load-bearing structural parts are a very difficult challenge, and many parts in the aviation sector do not pass certification tests [17,18,19,20,21,22]. Additive products have a wide range of utility, and structural and mechanical specifications. The structural and mechanical properties of the final components depend on many process parameters, including powder and its state (virgin or recycled) [23], build orientation, part position on the building platform, laser power and scanning speed, laser beam focusing, scanning strategy, powder layer thickness, and certain local geometrical parameters. Due to the usual position of the object, even with one set of process parameters, the conditions of powder melting, its solidification, and cooling rate may vary [5,10,22,24,25,26]. Casati et al. [27] reported reduced tensile strength with a wide scatter in the elongation to failure in LPBF AISI 316L stainless steel (316L SS) and attributed this to the presence of partially melted powder particles in the microstructure. AM components can also exhibit certain variations in mechanical and microstructural properties between different machines using the same set up, and even among the same parts printed in the same batch on the same platform [28,29].

As a result of the rapid heat-up, melting, and solidification (extremely high cooling rate of 10^3^–10^8^ K/s [30]) a fine microstructure with direction-dependent material properties is usually achieved [3,5,31]. The microstructure is very dissimilar to conventional casting/forging processes. Instead of polygonal equiaxed grains of conventional material, columnar anisotropic “weld-like” grains with an internal dendritic structure should be expected. These microstructural features have led to higher strength and lower ductility (elongation) when compared to conventional 316L SS in the vast majority of studies [7,10,11,32]. In this context, Tolosa et al. [33] found higher yield strength in 316L stainless steel samples processed by SLM in contrast to cast and rolled material, which can be attributed to the fine-grained microstructure. Thereby, elongation to fracture was not affected negatively.

Stainless steel AISI 316L is widely used in many industries such as aerospace, medical, automotive, and energy, both conventionally produced and printed as well. Due to high corrosion resistance, higher strength, biocompatibility, and easy post-processing 316L SS has been long used on orthopedic implants and prostheses [34]. Its low carbon content and fully austenitic structure limits the response of mechanical properties on thermal cycles and heat treatment, unlike the other additive materials such as 17-4 PH, Maraging 300, Inco718, and AlSi10Mg [35]. This limited sensitivity to heat treatment and its structural stability provides an advantage for further investigations, such as the impact of individual process parameters on the quality of components and the anisotropy of mechanical properties [10,27,33].

Considering these points, this study aimed to consider the effect of part orientation on anisotropy of mechanical properties and the microstructure of 316L SS. In addition to anisotropy, the influence of the process parameter “focus level” for two values was also evaluated. Analysis of mechanical properties was supported by optical microscopy and scanning electron microscopy for fracture surface investigation, providing an exceptionally comprehensive study. In addition, tendency to grain boundary formation during solution annealing was observed.

## 2. Materials and Methods

### 2.1. Stainless Steel CL20ES

Stainless steel powder labeled Concept Laser CL20ES (Concept Laser GmbH-GE Additive Company, Lichtenfels, Germany) is commercially available and designed for LaserCUSING LPBF additive machines of the same manufacturer. CL20ES is an austenitic stainless steel with chemical composition equivalent to W.Nr. 1.4404, AISI 316L. CL20ES is low carbon stainless steel suitable for welding and subsequent heat treatment due to sensitization resistance. The chemical composition of pre-alloyed CL20ES powder is given in Table 1. Table 2 provides a comparison of the mechanical and physical properties declared by the Concept Laser powder manufacturer and conventional steel manufacturer Cleveland-Cliffs Inc. (Cleveland, OH, USA).

### 2.2. Additive Machine and Processing Parameters

Test samples were built on a Concept Laser M2 Cusing additive manufacturing system (Concept Laser GmbH-GE Additive Company, Lichtenfels, Germany). M2 Cusing is a middle-class machine with a 250 × 250 × 280 mm build envelop. Test samples were sintered in two batches using a standard and verified set of parameters for CL20ES material recommended by the manufacturer and only virgin powder. The processing parameters used to prepare samples are as following: layer thickness—30 µm; powder size range—10–40 µm; exposition of plane (core)—200 W, 800 mm/s; exposition of contours—180 W, 1600 mm/s; beam diameter—0.15 mm; beam overlap—15%; chessboard pattern 5 × 5 mm, scan rotation between layers—90°.

Gór et al. [38] published a valuable critical review on the effects of process parameters on mechanical and microstructural properties, comparing also tensile strength properties for various settings (laser power; layer thickness; hatch spacing; scanning speed; energy density). The best results (UTS exceeding 600 MPa) were achieved with settings very similar to the recommended settings for CL20ES in three studies [39,40,41].

Parameter focus level was the only variable on the process parameters side: FL = −3 mm and FL = 0 mm. Parameter focus level (FL) describes the position (offset) of the laser beam focusing level with respect to the level of melting. In the case of CL20ES FL = −3 mm is used as a standard value.

Test samples were stress-relief-annealed immediately after printing in a Nabertherm N31/H (Nabertherm GmbH, Lilienthal, Germany) electric furnace without a protective atmosphere according to the following regime: heating to 550 °C in 3 h, dwell of 6 hours at 550 °C, slow cooling in furnace.

### 2.3. Test Campaign Samples

The scope of the experiment was designed to assess two main phenomena:Mechanical anisotropy in relation to the orientation of the test specimen;Effect of the “focus level (FL)” parameter on the mechanical properties and integrity of the samples.

Tensile test bars and V-notched bar impact testing samples were distributed on 2 test building platforms. (the first batch can be seen in Figure 1). The numbers of samples, sample orientation, and focus level parameter (FL) are listed in Table 3.

Tensile test specimen geometry fully respects gage section according to ASTM E8/E8M (Standard Test Methods for Tension Testing of Metallic Materials) [42]. The ASTM standard for the shape of test specimens was preferred to the ISO standard due to the reduction of neck cracking. The gage section of the sample had a diameter of 6 mm and length of five times the diameter. The central gage section was extended from 30 to 45 mm to have enough space for reflex strips of the laser extensometer. Tensile test bars were printed in three orientations: horizontal, −0°; tilted, −45°; and vertical, −90° (Figure 2).

V-notched bars were printed as 10 × 10 × 55 mm blocks with additional stock per surface. Test samples were subsequently ground (notch, functional surfaces) to meet dimensions, roughness, and GPS requirements of ASTM E23-07a specifications [43]. Charpy samples were first flat ground on a CNC grinder Mikronex BRH 20CNC (Mikronex, Prague, Czech Republic) using high pressure cooling, a 3M Cubitron II porous ceramic wheel, and very fine conditions. The notch was ground using oscillating grinding mode with high pressure coolant supply using a CBN wheel (Diatools 3V1/45° Progress PH C125 type; B107 abrasive, DEN9PH bond). Although CBN grinding is a significantly more expensive notch production technology (compared to milling), it offers the best ability to maintain the profile and minimizes surface hardening, residual stresses, and surface plastic deformation. V-notches were ground in three different positions with respect to the level of melting (see Figure 3):Type 1: VERTICAL BUILD; notch plane of symmetry identical with the level of melting.Type 2: HORIZONTAL BUILD; notch plane of symmetry perpendicular to the level of melting; the notch bottom lies in one level.Type 3: HORIZONTAL BUILD; notch plane of symmetry perpendicular to levels of melting; the notch bottom line crossing levels of melting.

### 2.4. Analytical Methods

The tensile tests were performed using LabTest 5100 SP1 servo-hydraulic high force universal testing equipment with a maximum load of 100 kN (Labortech LLC, Opava, Cyech Republic). Tests were carried out according to EN ISO 6892-1 [44] at room temperature using crosshead speed of 1 mm/min. Tensile strength (UTS (MPa)), yield strength (YTS (MPa)) and elongation (E (%)) were obtained, analyzed, and discussed. Five samples for each condition were tested. The same bars were also used for metallographic analysis. The fracture surface of tensile tests was inspected using a Jeol JSM-IT800 field emission scanning electron microscope (SEM) with an integrated EDS analyzer (Jeol Ltd., Akishima, Japan).

V-notched bars were tested on a Matest H060N (Matest S.p.A., Arcore, Italy) mechanical Charpy pendulum impact test device in accordance with ISO 148-1 [45] using 150 J and 300 J pendulum loads. Fracture surfaces were subsequently observed and measured under the microscope. Notch impact energy (absorbed) KV (J) was obtained, and notch impact toughness KCV (J/cm^2^) was calculated according to the formula KCV = KV/S0, where S0 is the minimum cross section under the notch. All tests were carried out at room temperature (20 °C).

Roughness and contour measurements were performed on a MahrSurf XCR 20 (Mahr GmbH, Göttingen, Germany) tactile contour measuring station. Straightness of the tensile test bar was investigated for vertical and 45° orientation tensile bars on an evaluation length of 50 mm.

With respect to anisotropy expected based on literature review, Brinell hardness was measured. Using load force of 187.5 kp (1839 N), a quenched ball of 2.5 mm in diameter was indented to the material (designated HBS 2.5/187.5 acc. to EN ISO 6506-1 [46]) on a Wolpert DiaTronic (Zwick & Co. KG. Wolpert, Germany) Two indentation directions were inspected, namely the vector of indentation parallel and perpendicular to the melting levels. Apart from macrohardness, microhardness mapping was performed on a FUTURE-TECH FM-100 auto-loading and auto-reading device (FUTURE-TECH CORP., Kawasaki-City, Japan). Measurements were done in line with EN ISO 6507-1 specifications [47], using 500 g loads (HV0.5) and a Vickers cone indenter. The indents were situated in 3 × 5 matrices.

Microstructural observation required preparation of metallographic samples. Hot mounted samples were ground using SiC abrasive of different FEPA grits in steps of 80, 120, 320, and 500, and polished diamond pastes of different particle sizes, namely 9 µm + rubber disc, 3 µm + fine cloths disc, and 1 um + fine cloths disc. A Keyence VHX 6000 (Keyence Corporation, Osaka, Japan) digital microscope with coaxial lighting was employed for polished and etched conditions of samples. Etching was performed electrolytically by 10% oxalic acid and the mixture HNO3 (50%) + ethanol (50%), which helps to suppress the internal grain microstructure and on the contrary highlight grain boundaries.

## 3. Results and Discussion

### 3.1. Anisotropy of Tensile Mechanical Properties

The scope of the experiment enabled anisotropy of the material in three strategic orientations to be evaluated: horizontal (0°), tilted (45°), and vertical (90°). In the majority of publications and materials of powder manufacturers’ datasheets, the mechanical values obtained from vertically printed test bars are given, so vertical test samples were considered as reference average values in this experiment as well—UTS = 607 ± 1MPa; YTS = 476 ± 20 MPa; E = 44 ± 2%. From the point of view of tensile strength properties, tilted 45° samples are more suitable and showed a 14% higher UTS (689 ± 2 MPa) and a 16% higher YTS (550 MPa). At the same time, a lower elongation was observed by relative decrease of 16% E (37 ± 2%). Horizontal samples exhibited better values in all tensile parameters, as UTS was higher by 10% (666 ± 1 MPa), YTS by 15% (548 ± 23 MPa), and E by 2% (45 ± 2%). In all orientations, lower scatter of UTS results were observed, while higher standard deviations were achieved for elongation to fracture. The results are clearly presented in Figure 4 and Table 4.

The experimentally measured values were partly in agreement with the already published studies [10,27,33], although the published results are not fully consistent with each other. Across publications and measured results, the highest UTS and YTS of the structure were proved in the direction parallel to the level of melting (horizontal bars, 0°), very closely followed by the UTS and YTS obtained by loading the structure at an angle of 45° to the melting levels (tilted, 45°). Published studies [7,10,27,33] also report significantly lower elongation to fracture E in the range of 14 to 43%, where lower values come mainly from older publications around the year 2010, for which there is a suspicion of higher porosity of test specimens. The authors of these studies also contradict the direction of highest elongation to fracture. Tolosa [33] denotes the direction of highest elongation perpendicular to the melting levels, while Casati [27] and Meier [10] denote the direction parallel to the melting levels.

Vertical samples made it possible to quantify the effect of the focus level parameter on tensile properties. Focusing the laser directly into the melting levels (FL = 0 mm) provided and indistinct increase of UTS by 1.2% (615 MPa) and a decrease of E by 1.25% (43%). The change in elongation was insignificant, while the yield strength showed a significant and repeatable improvement of 7.3% (511 MPa).

A comparison of the mechanical properties of AM produced AISI 316L and conventionally produced steel is often misleading, as it is very important whether typical or threshold values permitted by the EN 10088-3 specification [48] are used for comparison. It can be considered satisfactory that the strength properties in the as built state meet the EN 10088-3 specification for conventional solution annealed AISI 316L material. On the contrary, the values of elongation E and hardness HBS/HV are in the as built state at the limit values and cannot be repeatedly guaranteed regarding process parameters and spatial anisotropy of mechanical properties.

Austenitic stainless steels are characterized by a decrease of plasticity reserve, a decreasing elongation, and a hardness increase with increasing strength ratio (i = YTS/UTS), printed parts not excluded. The YTS/UTS ratio for conventionally produced and annealed parts ranges from approximately 0.4 to 0.55, depending on the level of strain hardening and dislocation density. For AM parts, the high density of dislocations is mainly caused by high cooling rates 10^3^–10^8^ K/s [11,30,31] and thermal stress in the structure. Depending on the orientation and process parameters, the YTS/UTS ratio in the range of 0.7 to 0.85 was found based on the performed experiment, while Tolosa et al. [33] reported values exceedingly even 0.9.

### 3.2. Notch Impact Toughness

Notch impact energy KV (J) and impact toughness KCV (J/cm^2^) were determined through a Charpy pendulum impact test (Figure 5). The worst result was achieved for the TYPE 1-vertical bar and notch plane of symmetry identical with level of melting (127 J/cm^2^). Two levels of melting were torn apart theoretically in this case, so reduced coherency and lower values were expected. Horizontal bars of TYPE 2 and TYPE 3 exhibited the same average impact toughness in both remaining notch orientations.

The influence of the process parameter focus level proved to be more significant against the influence of the position. The FL = 0 mm configuration on Type 1 notch bars parameter led to an increase in fracture toughness by 48% (188 J/cm^2^). Theoretically, this configuration should concentrate more energy directly into the melting level, which should result in a wider meltpool at the expense of the penetration depth and heating of the previous layers. Direct confirmation of this theory by measuring the meltpool width on metallographic samples was not reliable due to the overlap. For this purpose, it would be necessary to create and evaluate single laser track samples as published in [49,50].

AM manufactured samples exhibited excellent resistance to brittle or fast fracture across the experimental set of samples. The fracture surface had a ductile, dull, and fibrous appearance, without any bright regions. Side lips were significantly collapsed, and lateral contraction/extension could be observed. The room temperature was clearly far above the failure approach transition temperature (FATT) on the upper shelf of the S-curve.

Although the fracture was obviously ductile on the macroscopic scale in SEM observations, there was a mix of intergranular and transgranular mode on the fracture surface (Figure 6). Intergranular tear ridges were approximately the same size as the meltpool after laser passage. However, delamination at the layer level was not observed in any fracture surface orientation.

### 3.3. Hardness

Average Brinell hardness was almost identical in both measurement directions, parallel to melting levels –215 ± 3 and perpendicular to melting levels217 ± 11 HBS 2.5/187.5, respectively (see Figure 7). From the point of view of microhardness, no differences related to orientation were revealed. The FL = 0 mm configuration bars measured in vector of indentation parallel to melting levels reached average values of 244 ± 1 HBS 2.5/187.5 and exceeded the reference configuration, as could be expected based on the results of the tensile test. Hardness can be converted to Rockwell HRC hardness to be compared with powder manufacturers’ datasheets based on ASTM E140-97 [51].

The microhardness matrix was applied to metallographic sections to evaluate structural stability and hardness uniformity. Matrix 3 × 5 indents were defined, and outstanding measurement stability was achieved. Microhardness values at the edge of the solidification front of each linear laser passage and values inside the melted regions differed only minimally and unsystematically, regardless of both the FL parameter and orientation:FL −3 mm, indentation parallel to melting levels −230 ± 8 HV 05;FL −3 mm, indentation perpendicular to melting levels −229 ± 6 HV 05;FL 0 mm, indentation perpendicular to melting levels −235 ± 6 HV 05.

Considering the test load, such a low variation of measurement is difficult to achieve even for conventional material. The direction of the indentation vector relative to the melting level did not affect the measured values, while the FL = 0 mm configuration showed a slight increase at the microstructural level. All measurements showed a higher hardness in the as built condition compared to conventional AISI 316L material in solution annealed delivery conditions.

### 3.4. Fractography of Tensile Test Bars

Tensile tests were accompanied by fractography analysis using scanning electron microscopy (SEM). Fracture surfaces bore the marks typical of ductile materials: dimple morphology, significant necking phenomenon, and angle between the necked surface and axis of the test bar strongly distinct and close to 45°. The fracture itself was of a pure ductile nature. Figure 8 shows the overall fracture morphology. From the observations, metal with clean, matte, non-reflective, and highly rugged surfaces could be observed. Radial divisions of small and random pores could be observed on the fracture surface. On the surface of the sample, trapped but unmelted powder was visible, causing high surface roughness of AM parts (see Figure 9 and Figure 10).

Unmelted spherical powder particles could also be seen on the fracture surface (see Figure 11). The presence of these particles was more frequent near larger irregular shaped pores, defects, and their cavities. These cavities have their origin, for example, in lack of fusion, missing powder, craters, linear depression after recoater pass, or in insufficient melting of particle clusters. Irregular pores tend to collapse its shape and stretch during the tensile loading. On the fracture surface, a higher frequency of the presence of secondary cracks could be observed in these pores as initiators. The conformity of chemical composition of the particles and bulk material was confirmed by EDS analysis. The agreement of the chemical composition was also found by other authors, for example Diaz Vallejo et al. [52], who also determined the structure of the particles as purely austenitic on the basis of XRD pattern.

No obvious signs of oxidation were observed on powder particles by SEM or on particles trapped on metallographic sections, despite the less protective atmosphere generated by the nitrogen generator. These signs could probably be observed with repeated use of the powder. Pauzon et al. [53] described degradation of powder in the most common processing environment. The use of a nitrogen generator brings stable surface oxides on powder particles, which act as the inter-particle boundaries, resulting in oxide inclusions occurrence, higher oxygen pick-up, and inferior mechanical properties.

On the contrary, spherical pores did not exhibit this relation. The probable reason for this difference is that spherical pores are formed primarily in the melt pool by trapping a gas into the molten metal. Spherical pores on fracture surfaces were elongated (see Figure 12).

Figure 13 shows the dimple morphology of a metallic fracture at higher magnification. The equivalent diameter of dimples was below 1 um on average, that was lower than originally expected based on Margerit et al. [32]. This is also one of the reasons why there were no noticeable differences among fracture appearance among the samples.

### 3.5. Optical Microscopy and Porosity Analysis

Interpretation of the LPBF material structures is often challenging due to different appearances of the melting level section (top surface) and lateral sections as well as for unprecedented high melting and cooling conditions resulting in non-equilibrium structure condition. In addition, the lateral sections can differ from each other according to the angular rotation of the laser scanning direction. This can be neglected for the most common orthogonal rotation (90°). In any case, the rotation causes the successive layers to have differently oriented but periodically repeating melt pools.

Polished conditions of metallographic samples in both top and lateral sections revealed very low porosity levels. The very low frequency of spherical pores testifies to the suitable setting of power parameters with scanning speed. Suitable combinations of laser power and scanning speed to achieve low levels of porosity are well described in [52]. The largest spherical pores of 20 µm were found (excluded fracture surface, where pores are deformed). Observed porosities were lower than published in [5,54]. Non-spherical (irregular) pores distribution was very uneven. Very often, non-spherical pores occurred at sites of recoater blade damage or sites of problematic recoating due to the presence of powder agglomerates (see Figure 14). For this mechanism of pore formation, linear groups in the direction of powder application are typical.

Non-spherical pores often retain unmelted or partially melted particles in their cavities. By comparing Figure 9 and Figure 15, we can observe the same structure. On these particles, pure internal dendritic solidification structures, which Casati [27] and Röttger [31] revealed on globular surfaces, were seen.

Porosity analyses were performed using Keyence analytical software on three top sections in different heights of spare tensile test bars (top, central, bottom = close to the building platform). The area of 9.72 mm^2^ was in the central part. Pores and voids were sorted by size to create a distribution curve as shown in Figure 16 and Table 5. The highest porosity of 0.103% was detected on the top section due to the large number of counts with small diameters. The central part exhibited 0.036%, and bottom section contained 0.089% of porosity. At the same time, a more frequent occurrence of larger pores could be observed in the lower layer, including the largest observed non-spherical pore.

The obtained results are not entirely consistent with expectations based on process knowledge. As the built progresses layer by layer, the rubber blade of the recoater wears, which usually leads to uneven application, scratches in the applied powder surface, and the formation of larger non-spherical pores, which were not present in this case. Although there was an overall higher rate of porosity in top sections, small sized spherical pores predominated.

Etched micrographs at lower magnification showed solidification tracks created after individual passes of the laser beam. Laser tracks are displayed in Figure 17 as the pattern of linear weld beads in the melting levels. Due to misalignment of the section with melting level and the partial re-melting, the previous 90 degree rotated laser scanning direction could also be observed.

The lateral section was composed of hatch overlapped melt tracks of semi-circular shape that represented melt pool cross-sections with very clear melt pool boundaries. Melt pool width and depth ranged between 90 and 150 µm. Across melt pools, epitaxial growth driven by heat flow in the build up direction appeared on a macro-scale. The reason for that growth was partial remelting of previous layers.

Based on the analysis of metallographic sections in high magnification on SEM, the authors in [3,27,54] described a cellular and columnar internal solidification submicrostructure. This submicrostructure could also be observed via light optical microscopy (see Figure 18). Cellular submicrostructures significantly predominated on lateral sections, while both types could be randomly observed on melting level sections.

### 3.6. Annealing and Recrystallization, Grain Formation

In the previous section, a typical inconsistency of the macrostructure of the LPBF manufactured material and anisotropy of mechanical properties were found. This anisotropy persisted even after the recommended stress relief annealing (550 °C/6 h). The structure retained a fine solidification character, and no grain growth occurred.

For comparison, the damaged samples after the tensile test were used for subsequent solution annealing according to the regime (1070 °C, air furnace, warming up for 10 min in heated furnace, holding time 10 min, rapid water cooling). The subject of interest included two areas:(a)The clamping part not subjected to previous deformation (Figure 19);(b)The proximity of the fracture surface with maximum deformation (Figure 20).

In the as built + stress relief condition, a considerable elongation of the melt pools at the point of failure was evident, which was in accordance with the high elongation measured during the tensile test. After solution annealing, full polygonization of the grain was found, and in these specific lateral sections, the microstructure was practically indistinguishable from conventional material. Annealing twins appeared with great frequency in both areas. The area near the fracture surface had a significantly higher deformation energy and higher density of dislocations, which was the driving force of recrystallization and nucleation of new grains. Grains in the range of ASTM number G8 to G10 were found near the fracture surface and G4 to G7 in undeformed areas, according to ASTM E 112 specifications. According to the Hall–Petch relationship, a growth in grain size contributes to the significantly lower yield and tensile strength. Based on the Hall–Petch relationship, lower yield strengths can be expected for recrystallized material. Significant migration of meltpool boundaries to polygonized grain boundaries in a relatively short time of 10 minutes indicates high recrystallization kinetics. It is highly probable that the steel will have very good technological properties during soldering and brazing.

Fine-grained carbides were evenly distributed in the structure. Precipitation of carbides at the boundaries and sensitization of the microstructure to intergranular corrosion were not found. In both areas, the annealing twins were clearly visible, as in conventionally-produced material (see Figure 21 and Figure 22). Despite even distribution, Benarji [55] described deterioration in corrosion resistance after recrystallization annealing.

## 4. Conclusions

The purpose of the study was to analyze mechanical and microstructural properties of AISI 316L (W.Nr. 1.4404, CL20ES) manufactured using LPBF additive technology. In addition to a complete analysis of samples made with standard process parameters, the effect of focusing level shifting to the melting level was evaluated, and differences from baseline were reported. In the case of mechanical properties, vertical test samples loaded in the building direction and printed with standard parameters were considered as baseline (UTS = 607 MPa; YTS = 476 MPa; E = 44%). The findings led to the following conclusions:The anisotropy of tensile properties persisted in as-built material after stress relief annealing. Tilted (45°) samples showed 14% higher Rm, 16% higher YTS, and 16% lower E. Horizontal samples exhibited the best tensile properties—10% higher UTS, 15% higher YTS, 2% higher E against baseline.The shifting of focus level to the level of melting (FL = 0mm) brought an indistinct increase of Rm by 1.2% and a 1.25% increase of E, but a significant increase of YTS by 7.3%.Depending on the orientation and process parameters, the YTS/UTS ratio in the range of 0.7 to 0.85 was found.Although the effect of notch orientation to melting levels was not significant based on the results, the lowest values of impact toughness KCV (127 J/cm^2^) were achieved in the case of orienting the notch plane of symmetry into the melting level. Theoretically, two subsequent melting levels were torn apart in this case. Considering real coherency between melting levels, results were in line with expectations.The impact toughness KCV of FL = 0 mm vertical samples was higher by 48% (188 J/cm^2^) compared to baseline.Macrohardness measurement represented by Brinell scale did not exhibit any systematic differences among planes of measurement as well as among vectors of indentation (215–217 HBS 2.5/187.5). From this point of view, consistent results of macrohardness and microhardness (229–230 HV0.5) could be observed.Using both hardness measurement methods, F= 0 mm samples exhibited higher values (244 HBS 2.5/187.5, resp. 235 HV0.5).Fractography analysis was performed using both scanning electron and optical microscopy. Fracture surfaces of tensile test samples contained typical features of ductile cracking with significant necking, high angled neck surface, and fine dimple morphology of the fracture. Several defects were observed on the metal-clean fracture surfaces—secondary cracks, voids, deformed spherical pores, non-spherical pores, and unmelted powder particles.Porosity investigation was performed in three levels along vertical tensile test samples with very promising results in the range of 0.036% to 0.103%. Across all samples according to the pore size distribution curve, more then 95% of total pore area was created by spherical pores smaller than 25 µm.Microstructural investigation of solution annealed (1070 °C) tensile test samples showed outstanding tendency to recrystallization, grain polygonization, annealing twins formation, and even distribution of carbides in solid solution.

As mentioned above, the vast majority of studies [38,39,40,41,52] focus on the process parameters of the basic setup and their derived indicators: laser power; layer thickness; hatch spacing; scanning speed; and energy density. The presented experiment has a completely different concept and deals with a focus level parameter that is not usually considered a process parameter, despite the fact that it obviously affects mechanical properties across combinations of these common basic process parameters. Focus level can be intentionally changed in the service settings of the machine (presented experiment) or unintentionally by incorrect installation during service of hardware. The level shift can also appear to a limited extent due to wear and frequent use, and therefore the machine should be inspected periodically to avoid these alternations.

In addition to a comprehensive description of the anisotropy of mechanical properties, the presented article indicates the influence of the focus level parameter on the mechanical properties and the influence of plastic deformation on recrystallization processes and grain growth. Both of these influences have an indicative character, and in the further development it would be very appropriate to quantify these phenomena in several levels. At the same time, it would be good to exclude the persistence of anisotropy of mechanical properties in solution annealed conditions.

LPBF technology is revolutionary and undoubtedly belongs to the fastest growing industries. The performed analysis confirms the high quality of additively manufactured material, which can be compared with conventionally produced material. In the development phase of new products, it is necessary to consider real mechanical properties and their anisotropy, as well as their modification through process parameters. The situation is even more serious because there is a tendency to use additive manufacturing for topological optimization tasks and the creation of lightweight structures. It is detailed knowledge of mechanical properties and its anisotropy that opens the possibilities of advanced topological optimization in relation to the production process and part orientation.

## Figures and Tables

**Figure 1 materials-15-00551-f001:**
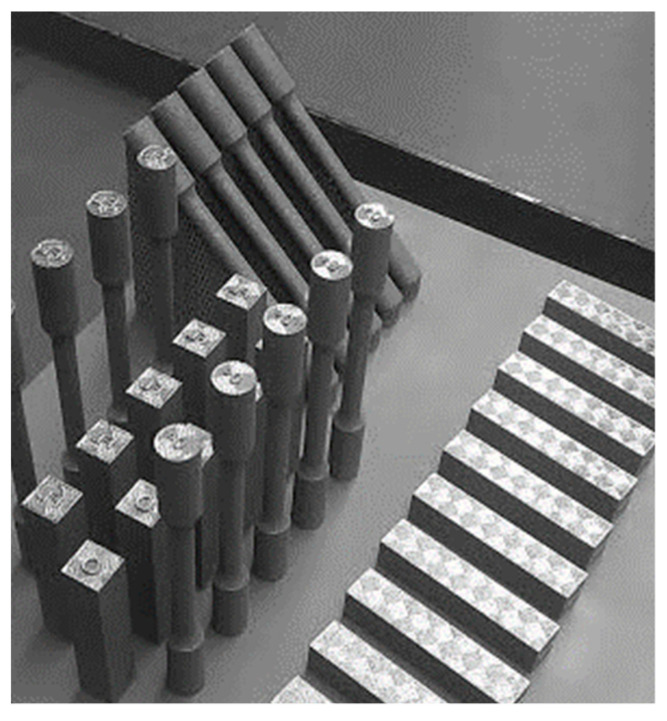
First batch of samples.

**Figure 2 materials-15-00551-f002:**
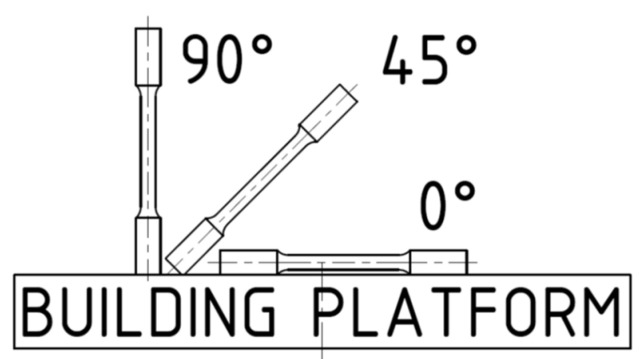
Tensile test bar orientation.

**Figure 3 materials-15-00551-f003:**
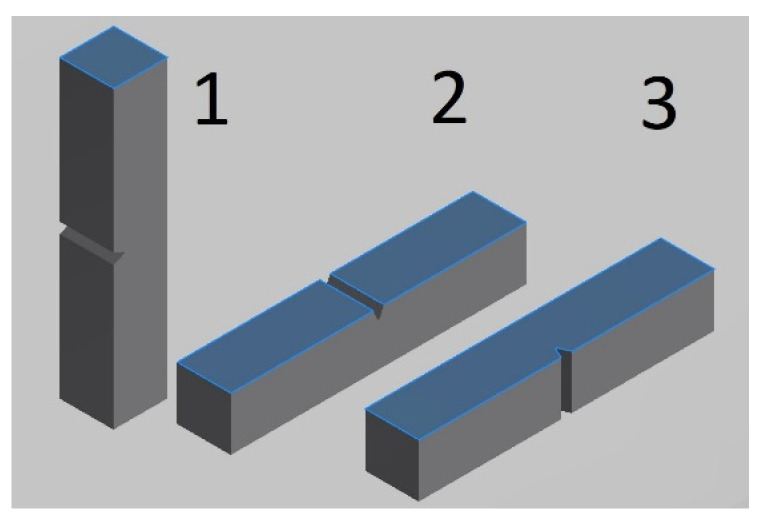
V-notched bars—building direction, notch orientation, levels of melting (blue color).

**Figure 4 materials-15-00551-f004:**
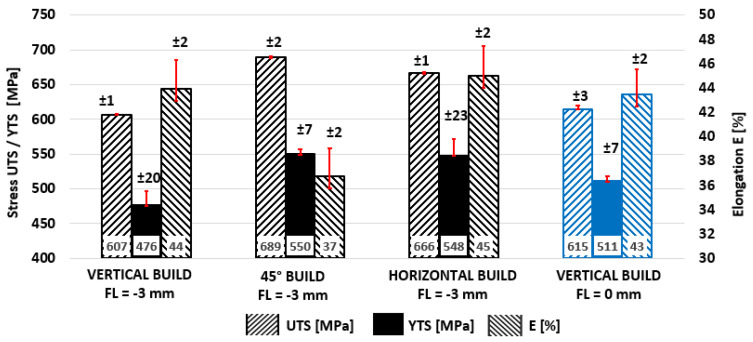
Tensile strength properties for the AISI 316L (CL20ES) specimen.

**Figure 5 materials-15-00551-f005:**
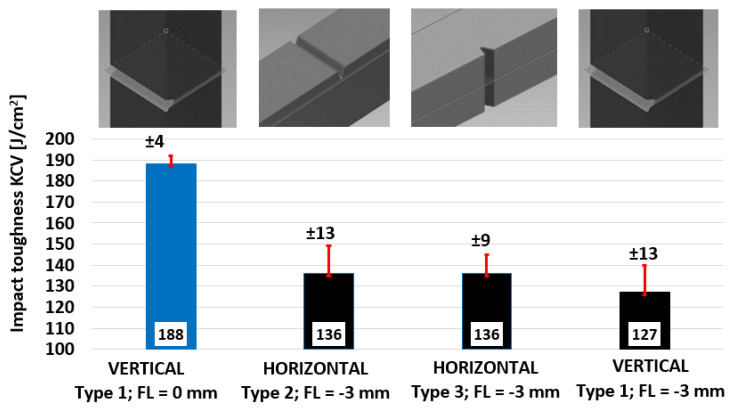
Notch impact toughness for the AISI 316L (CL20ES) specimen.

**Figure 6 materials-15-00551-f006:**
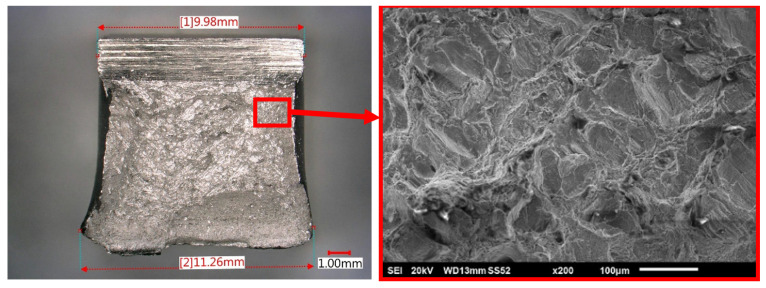
Fractured surface of notch impact bars: ductile appearance and mix of inter/transgranular mode (SEM).

**Figure 7 materials-15-00551-f007:**
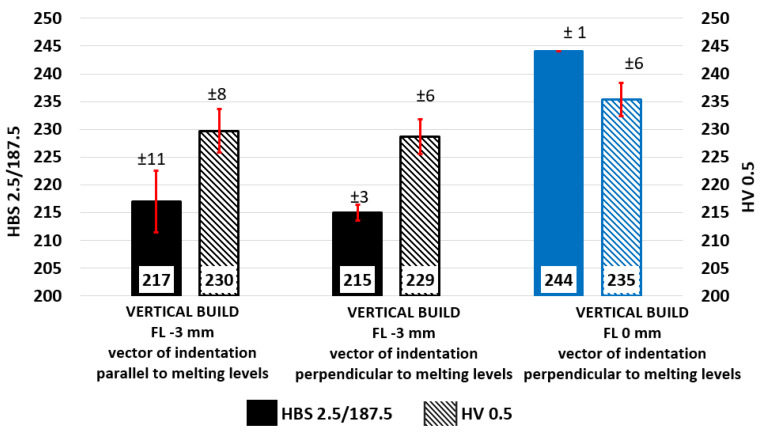
Brinell hardness and Vickers microhardness for the AISI 316L (CL20ES) specimen.

**Figure 8 materials-15-00551-f008:**
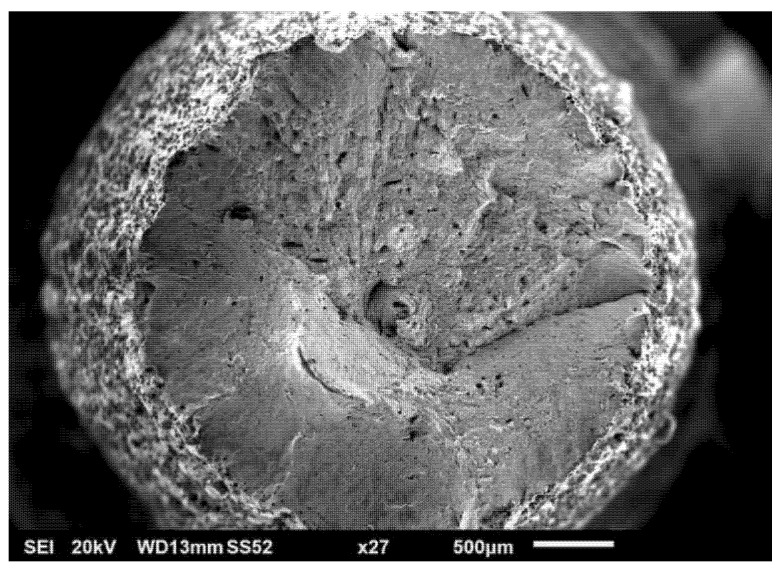
SEM observation, secondary electron spectrum—overall fracture morphology.

**Figure 9 materials-15-00551-f009:**
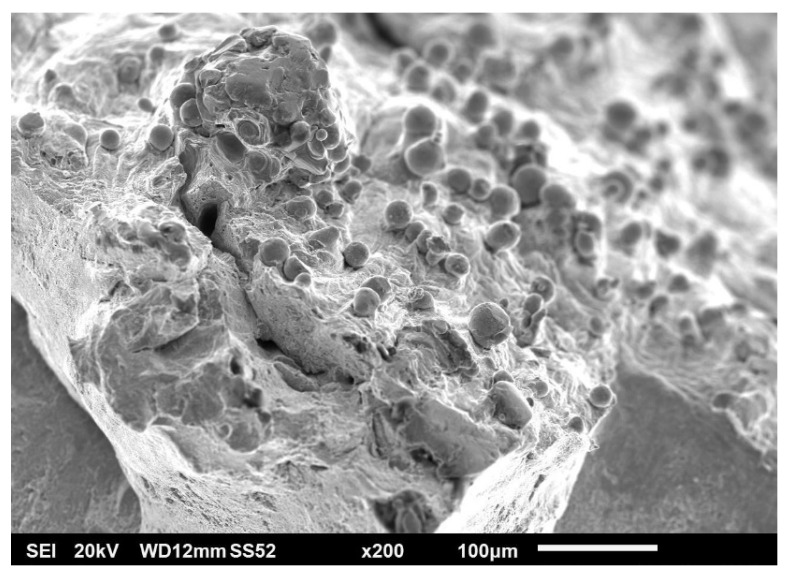
Unmelted powder trapped on outer skin in close proximity of fracture surface (SEM).

**Figure 10 materials-15-00551-f010:**
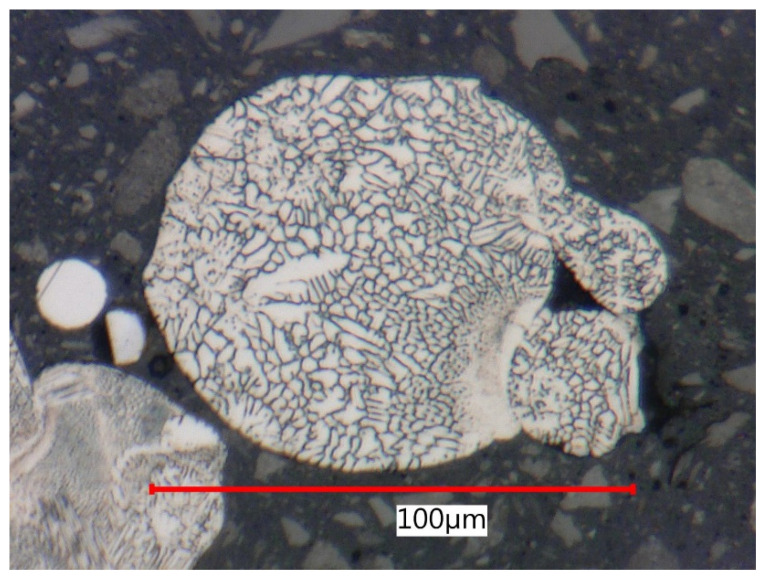
Large unmelted powder agglomerate trapped on outer skin—fine rapidly solidified dendritic structure.

**Figure 11 materials-15-00551-f011:**
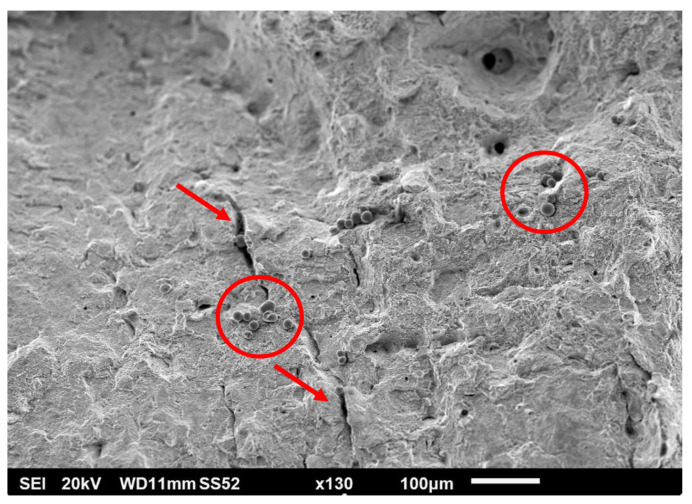
Radial secondary cracks—marked by arrows and unmelted particles on fracture surface—in red circles.

**Figure 12 materials-15-00551-f012:**
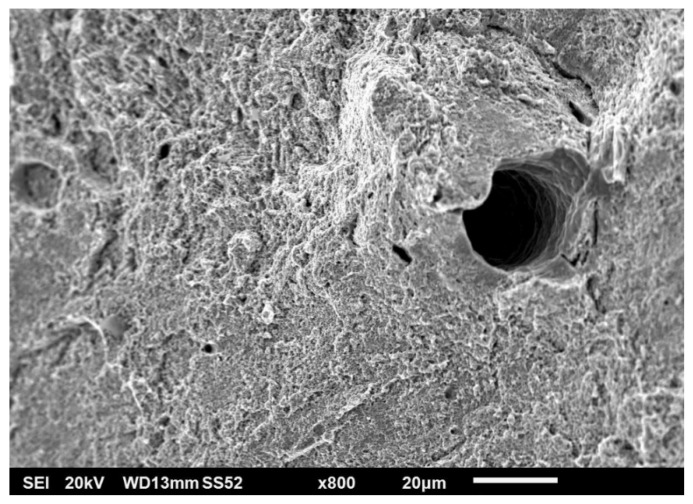
Large spherical pore elongated in direction of loading; surrounded by dimple morphology.

**Figure 13 materials-15-00551-f013:**
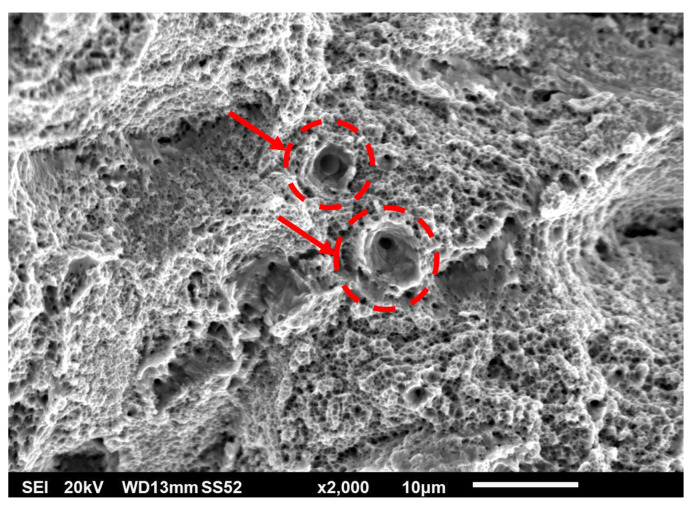
Dimple morphology of fracture surface; wide craters around two small pores—in red circles

**Figure 14 materials-15-00551-f014:**
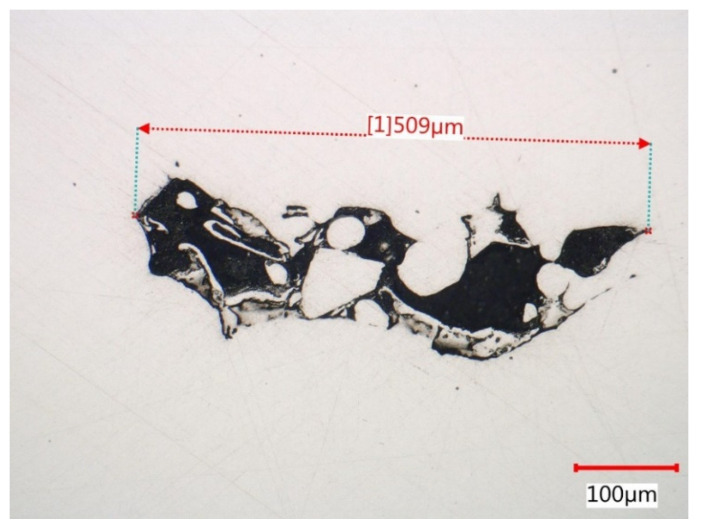
Non-spherical pores filled with unmelted powder particles.

**Figure 15 materials-15-00551-f015:**
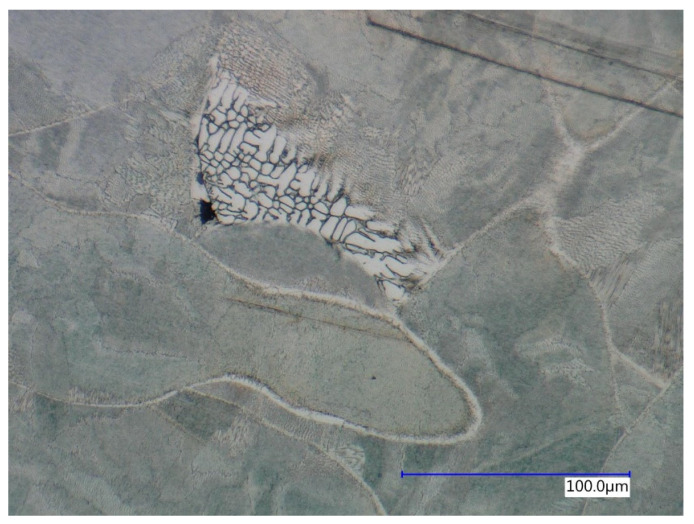
Dendritic solidification structure of unmelted particle.

**Figure 16 materials-15-00551-f016:**
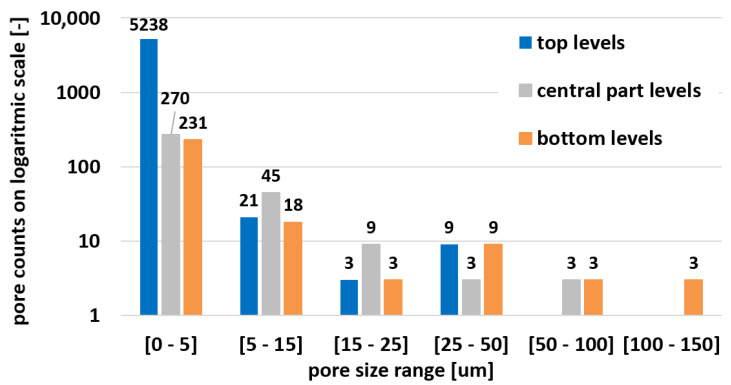
Pore size distribution.

**Figure 17 materials-15-00551-f017:**
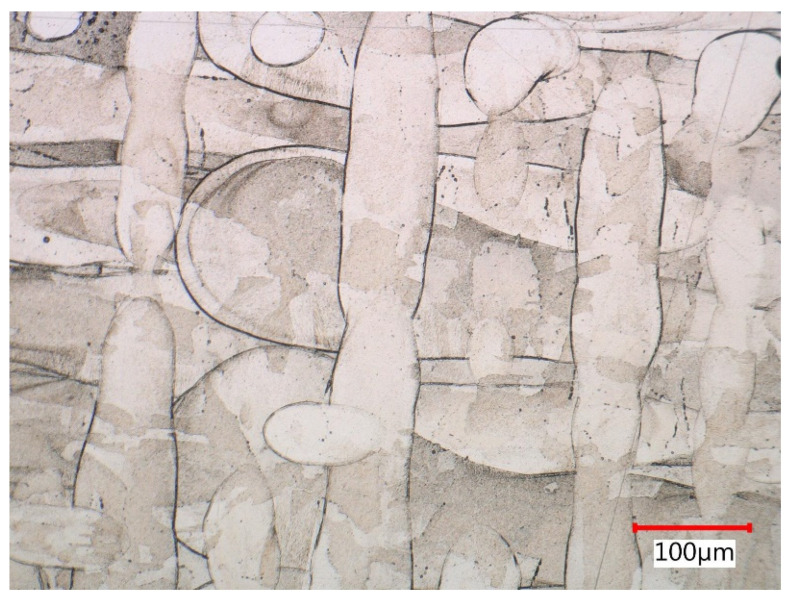
Solidification tracks in melting levels (oxalic etchant).

**Figure 18 materials-15-00551-f018:**
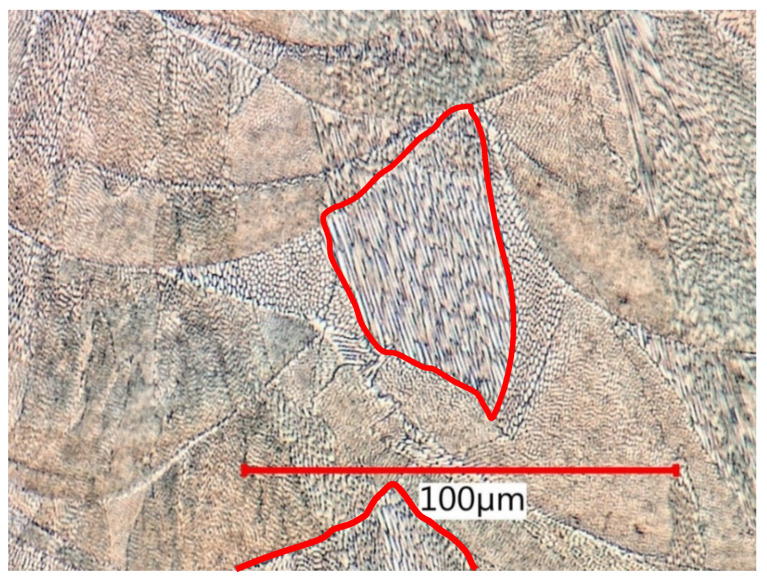
Predominating cellular solidification submicrostructure on lateral section surrounding columnar regions (red loops).

**Figure 19 materials-15-00551-f019:**
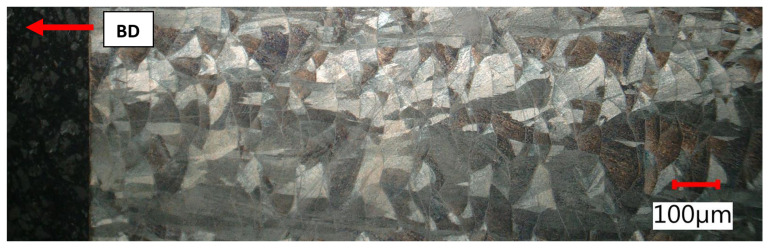
Undeformed clamping part area, as built + stress relief, lateral section.

**Figure 20 materials-15-00551-f020:**
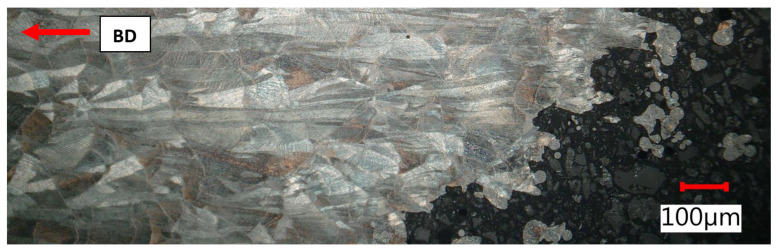
Elongated meltpools in close proximity of tensile fracture, as built + stress relief, lateral section.

**Figure 21 materials-15-00551-f021:**
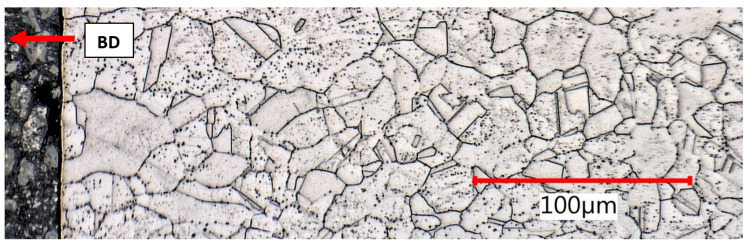
Effect of solution annealing in the clamping part area—G4 to G7 grain size.

**Figure 22 materials-15-00551-f022:**
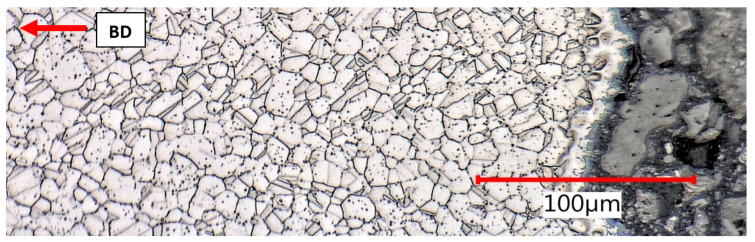
Effect of solution annealing in the proximity of tensile fracture—G8 to G10 grain size.

**Table 1 materials-15-00551-t001:** Composition of CL20ES powder and conventional 316L (declared) [36,37].

(w.t.%)	Fe	Cr	Ni	Mo	Mn	Si	P	S	C
CL20ES	Balance	16.5–18.5	10–13	2–2.25	0–2	0–1	0–0.045	0–0.03	0–0.03
316L		16–18	10–14	2–3	0–2	0–0.075	0–0.045	0–0.03	0–0.03

**Table 2 materials-15-00551-t002:** Mechanical properties of CL20ES and conventional 316L (declared).

Property	CL20ES Data Sheet [36]	Convent. 316L [37]
Yield point YTS (MPa)	^(1)^ 374	290
Tensile Strength UTS (MPa)	^(1)^ 650	627
Elongation A (%)	^(1), (2)^ 65	55
Young’s modulus E (MPa)	^(3)^ approx 200 × 10^3^	193 × 10^3^
Thermal conductivity λ (W/mK)	^(3)^ approx 15	16
Hardness	^(4)^ 20 HRC	79 HRB

^(1)^ Tensile test at 20 °C according to DIN EN 50,125 [36]; ^(2)^ by using a special heat treatment a higher elongation can be achieved [36]; ^(3)^ specification according to the material manufacturer’s data sheet [36]; ^(4)^ hardness test according to DIN EN ISO 6508 [36].

**Table 3 materials-15-00551-t003:** The scope of samples.

Orientation	Type	Focus Level (mm)	Quantity
Vertical	Tensile	−3	5
Horizontal	Tensile	−3	5
45°	Tensile	−3	5
Vertical	Tensile	0	5
Vertical	Notched	−3	5
Horizontal	Notched	−3	10
Vertical	Notched	0	5

**Table 4 materials-15-00551-t004:** Summary of measured data, typical conventional material, and EN 10088-3 specification [36,37,48].

	YTS (MPa)	UTS (MPa)	E (%)	YTS/UTS Ratio	HBS	HV
CL 20 ES Datasheet	374	650	65	0.58	-	-
316L typical (+AT)	290	630	55	0.46	180	180
316L acc. to 10088-3	min 200	min 500	min 45	0.4	max 215	max 226
VERTICAL 90°, FL -3	476	607	46	0.78	217	230
TILTED 45°, FL-3	550	689	37	0.8	217	230
HORIZONTAL 0°, FL-3	548	666	45	0.82	217	230
VERTICAL 90°, FL 0	511	615	43	0.83	244	256

**Table 5 materials-15-00551-t005:** Pore size distribution count.

Max. ⌀ (µm)	(0–5)	(5–15)	(15–25)	(25–50)	(50–100)	(100–150)	Pore Area Summary (mm^2^)	Porosity (%)	Maximum Size (µm)
Top	5238	21	3	9	0	0	9981	0.103	43
Central	270	45	9	3	3	0	3527	0.036	55.8
Bottom	231	18	3	9	3	3	8688	0.089	119.4

## Data Availability

Data are contained within the article.

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
