# Peer review of "Mechanical and Microstructural Anisotropy of Laser Powder Bed Fusion 316L Stainless Steel"

_materials, 2022, doi:10.3390/ma15020551_

Round 1
Reviewer 1 Report
The subject of the article is very interesting and has a great contribution to the development of modern manufacturing technologies such as addtive technology. The conducted analysis of the material sintered from 316L stainless steel powder significantly broadens the scope of knowledge regarding the structure of this material. Reviewed article is very interesting and write at good scientific level. However, the manuscript requires a revision prior to publication.
The following suggestions have to be addressed before publication of the paper:
- In Figures 4, 5, 7, it would be worth adding the standard deviation values.
- Please check the values, because in a lot of values occur comma instead of dot.
- In article is mentioned „The conformity of chemical composition of the particles and bulk material was confirmed by EDS analysis”, please present the results analysis of EDS.
- In Conclusion please indicate how the presented results analysis affects on the current knowledge in range of additive manufacturing, and also please indicate further directionn of experimental research.
Author Response
Dear Reviewer,
I really appreciate the time you spent reviewing my article and thank you for your constructive comments and helpful reviews. I have incorporated the individual comments into the text of the manuscript. Let me make a brief statement on your points:
- It would certainly be appropriate to add standard deviations and not lose clarity in the figures, that is a bit challenge.
- Yes, I put a decimal comma almost everywhere. I'll replace with a dot.
- I did not present the EDS analysis in the article, as we did not want to publish it in detail. We repeatedly achieved conformity of chemical composition, with no oxygen or nitrogen enrichment. And the aim of the analysis was to exclude external contamination at that moment, to be honest. The only evidence, that the analysis was performed are pictures with points of interest from Jeol SEM microscope. We still keep the samples and we could have the analysis done again, but not within ten days.
- Some conclusions are followed by other publications by other authors, especially in the field of anisotropy of mechanical properties. It was very meaningful to conduct a new and comprehensive study of the influence of the FL parameter on quality and mechanical properties. In the present article, the choice of the two levels is only indicative,
Thank You, Yours sincerely
Zdeněk Pitrmuc

Reviewer 2 Report
It will be necessary/better to take into account: 1. difference and importance of primary Cristal /solidification structure and microstructure after heat treatment. It should be clarified. 2. The influence of Oxigen/Nitrogen content and Non metallic Inclusions as well on the investigated properties. It can be added. It can be the object to the next investigation.Author Response
Dear Reviewer,
I really appreciate the time you spent reviewing my article and thank you for your constructive comments and helpful reviews. I have incorporated the individual comments into the text of the article. Let me make a brief statement on your points:
- Primary solidification structure and polygonal recrystalised structure will be described more clearly.
- The effect of nitrogen and oxygen on tensile properties is very limited in case of AISI 316L, compared to reactive AM materials. Department of Industrial and Materials Science, Göteborg, Sweden has been working in this area for a long time. Pauzon et al. did not reveal significant differences in mechanical properties and microstructure using argon 5.0, Nitrogen 5.0 and nitrogen from internal generator (our case).
Pauzon, C., Hryha, E., Forêt, P., & Nyborg, L. (2019). Effect of argon and nitrogen atmospheres on the properties of stainless steel 316 L parts produced by laser-powder bed fusion. Materials & Design, 179, 107873. https://doi.org/10.1016/j.matdes.2019.107873
The presence of non-metallic inclusions is very limited when using virgin powder. Contamination is very unlikely and at the same time a source of oxides on the surface of “re-used” powder particles cannot be expected. If this already happens, a negative effect on the mechanical properties can be expected. The formation of oxide inclusions after subsequent heat treatment is described by Saboori et al.
Saboori, A., Aversa, A., Marchese, G., Biamino, S., Lombardi, M., & Fino, P. (2020). Microstructure and Mechanical Properties of AISI 316L Produced by Directed Energy Deposition-Based Additive Manufacturing: A Review. Applied Sciences, 10(9), 3310. https://doi.org/10.3390/app10093310
Thank You, Yours sincerely
Zdenek Pitrmuc

Reviewer 3 Report
In this paper, the authors explore the mechanical and microstructural anisotopy of LBF 316L stainless steel. The paper itself is very well-presented, but the study is small compared to many similar ones in the past. This exact type of study has been done hundreds of times in the past 10 years and 316L is one of the most common types of materials used. The authors themselves cite some of the review papers done on this exact topic, since there are so many previous (nearly identical) studies. The authors need to make a better argument for novelty and demonstrate what is new and useful about this study.
As the paper currently sits, I do not see any major novelty or engineering literature contribution in publishing it in Materials. Perhaps a major a revision by the authors can improve it and make it suitable for publication. However, I am not confident this can reasonably be done in the few weeks normally given for major revisions. Therefore, I recommend the paper be declined but the authors encouraged to resubmit to Materials after they have had time to thoroughly address the issues with the paper (and probably add a significant amount of new material in Section 1 and Section 3 at least).
Some specific comments:
1. It is my understanding reading the paper that the Charpy samples were printed with the notches already cut and they were simply polished later to meet the standard that was being followed. Is this the case? If so, it needs to be clearly stated. In impact testing, the notch is the most important factor for repeatability and so most or all standards require the notch to be machined or cut (not ground, since this can heat the notch surface too much) to an exact specification as the last step before testing. The only time when a notch should be printed with a sample is if the point of the study is to observe notch influence. In all other cases, it violates the common standards and can introduce unnecessary error into the experiment.
2. How and how long were the samples conditioned before testing, especially the impact testing samples?
3. The literature review for this paper is very weak and should be improved. The review only covered a very small sample of the many similar studies, which should be improved if the authors are trying to show novelty with this study. This should be the first item the authors try to address, since a strong literature review will show a good path forward for the authors to make this paper publishable.
4. There is very little discussion of the results and their implications on materials, design, etc. Since this topic is so widely studied and so many papers have been published, a simple property collection and description of the fracture surfaces is not enough for a new research article in this area. The authors should review some recent papers published on this topic in Materials to get an idea of what is expected for publication. Most of the newer papers are far longer and more extensive than this one and all have clear novel contributions toward materials design, engineering materials, fracture mechanics, etc.
5. The final paragraph of Section 4 would have been a true perhaps 10 years ago, but LPBF is now a very well-established AM process and have been already applied to end-user manufacturing. It is not revolutionary and has some serious limitations, as well as a high cost. It will continue to grow and refine, but it really cannot be considered an emerging or futuristic process at its current level of development.
Please note that these comments are only to provide helpful feedback for the authors and preserve the integrity of the engineering/materials literature. Nothing said here is aimed at the authors personally nor should be taken as questioning their abilities or research skills.
Author Response
Dear Reviewer,
I really appreciate the time you spent reviewing my article. Regardless of the negative evaluation, thank you for your constructive comments. Since the remaining reviews are mostly positive, I will try to improve the article and incorporate also your recommendations. Let me make a brief statement to specific comments bellow.
Yes, 316L steel is by far the most common and researched material. However, this material was chosen intentionally due to the possibility to compare alternative processing parameters under own standard parameters and the results of other researchers. I consider the presentation of mechanical properties for focus move FL=0 mm (identical focusing/melting level) to be an unexplored area. We also consider the presentation of the significant influence of the previous plastic deformation on the formation of grain boundaries during recrystallization as a unique output.
The method of manufacturing Charpy samples was performed very precisely. The samples were printed without notches and with additional stock per surface. The risks of overburning during grinding are very well known to us. We have been dealing with surface integrity after finishing operations for a long time. We are familiar with the production standards of aircraft engine manufacturers GE (P1TF79) and Rolls Royce (MSRR9968) and we provide the production of test specimens commercially. Charpy samples were first flat ground on a CNC grinder using high pressure cooling, a 3M Cubitron II porous ceramic wheel and very fine conditions. The notch was ground using oscillating grinding mode, with high pressure coolant supply, CBN wheel (Diatools 3V1/45° PROGRESS PH, B107, DEN9PH, C125). This tool allows the best ability to keep the profile and absolutely minimal impact on the surface. This is the most expensive and most suitable way to produce the notch. Compared to milling, it has a very low load on the surface, which prevents the formation of deformed layer and prevents the introduction of residual stresses as well.
The test was performed at room laboratory temperature, which is maintained between 19 and 21 ° C. The samples were also stored in the laboratory, so they were temperature stabilized.
Literary review certainly does not and may not list all available sources, however the review involves leading journals such as Additive Manufacturing or Materials & Design. I do not know how to approach this problem regarding positive evaluation of remaining reviewers.
Yes, the final paragraph may sound like a cliché, but it is certainly not contrary to reality. The final paragraph should not give the impression that the authors consider the LPBF to be a futuristic technology. However, it is true that 3D printed products have major limitations in aviation and many other critical industries. There are still problems with process control, production stability and final parts inspection methodology. LPBF is undoubtedly very well established in specific less demanding applications such as mould making or dental implants as mentioned in the introduction.
Thank You, Yours sincerely
Zdenek Pitrmuc

Reviewer 4 Report
- Although there are a lot of data in the open literature on the microstructure and properties of SS316L produced by PBF, this work particularly highlighted the anisotropy effects on the mechanical properties.
- The work contains valuable information and results; however, few results lacked adequate discussions. My comments in the manuscript have highlighted some of my concerns.
- The introduction is overly lengthy. Some of the information can be removed as most interested readers will already be familiar with PBF-L.
- In some parts of the manuscript, the sentences are difficult to understand. Therefore, I have added my comments where applicable for modification.

Author Response
Dear Reviewer,
Thank you very much for the factual evaluation with specific suggestions for correction. A PDF file with comments is the best way to review. I tried to incorporate the vast majority of your comments. Let me comment only problematic ones, others were implemented:
- Use of SI units: I changed Rm, Re to UTS/YTS. Units from N/mm2 to MPa as it appears in other MDPI papers. Charpy Impact toughness has corect units I think. I am using calculated toughness (related to sample size) KCV [J/cm2]. I do not use absorbed impact energy KV [J] intentionally despite knowing it is more frequent in the papers. I hope this is a good approach. Using base or derived units would be really unusual (stress… kg⋅m−1⋅s−2, N/m2).
- Figure 12: Yes deformation bands (I mean signs of deformation after tensile loading) are not visible enough in lower magnification. Pores are really elongated in the direction of loading that could be noticeable while scanning the surface (maybe even on high magnification optical microscope or confocal it could be better proved).
Thank You, Yours sincerely
Zdeněk Pitrmuc

Round 2
Reviewer 3 Report
To the editor: Thank you for granting me more time to review the revised version of this manuscript. The revision is very poorly done and does not really address any of my concerns with the work.
To the authors: Perhaps you submitted the wrong version of the manuscript for the revision but there is almost no change the manuscript despite it needing a significant amount of revisions before it could be suitable for publication. The comment-based highlighting method used by the authors is also extremely confusing and it was very difficult to find the changes made by the authors. In the future, please simply make changes and highlight them with a color, pointing out line numbers of changes in your reply letter. Some specific points:
1. The authors do have good technical answers for some of my questions about the tests and sample preparation in the reply letter, but these answers do not seem to have made it into the revised manuscript. The credibility of the study depends largely on the credibility of the experimental setup, so I do not understand why the authors did not add these details even after they were specifically requested by the reviewer. I read the revised paper twice on different days and did a keyword search and did not find that they were anywhere in the revised paper. There is no question that these experimental details are necessary for the paper.
2. The authors brushed off my serious comment about the literature review. This study is weak (even after revision) and has no obvious novelty (since this exact study has been done so many times, as early as 2008), so it is necessary for the authors to demonstrate its need in the literature by pointing to a specific hole or need in the literature. There is already more than enough studies on this exact question with these exact parameters, etc. and so this study appears to be a re-replication (something like the 8th or 10th replication). This is fine but not really something that is appropriate for a major international journal. I suggested improving the literature review as a path to making the study at least claim/demonstrate some novelty so it is worth adding to the body of literature for the field. The purpose of a review is to show novelty and the need for the work, not just to show some representative sampling of past literature in "leading journals".
3. The authors still did not really add any discussion of the results or implications from this work, so it is still basically just a presentation of results, none of which are new to the literature. This was another area where the authors could have argued that their study was something valuable for the literature but declined to do so.
In conclusion, and after a lot of thought and reading the revised article carefully, this work is not suitable for publication even after revision. The authors were asked to complete a major revision and did not make a serious attempt to do so, brushing off or ignoring most of the issues brought up. Therefore, I cannot support the publication of this work and recommend it not be given further consideration by the editors of Materials.
Author Response
Dear Reviewer,
Let me respond to your comments from the second round of the review. In the first round, I responded to specific comments from all reviewers. There was agreement on some of the comments among the opponents, and it was possible to react clearly to them in the article. Much of your comment was evaluative, not specific incentives or requirements for correction. I therefore had a limited opportunity to modify the article and I decided to involve my explanation to answer. It wasn't about ignoring your comments at all, and I really appreciate your feedback. It is especially difficult for me to react to comments where you do not agree with other reviewers - such as the evaluation of the literature review, presentation of results and conclusion.
I checked the uploaded file and all the revisions to the manuscript were marked up using the “Track Changes” function as it is required by the editor. Please see printscreened pictures on the end of this letter. All the altered passages are very well visible.
- Technical answers regarding test samples and experimental setup that I was explaining in my first letter were now incorporated in the article based on Your requirements. (lines 185-195)
- The vast majority of studies focus on the process parameters of the basic setup and their derived indicators – Laser Power (W); Layer Thickness (um); Hatch Spacing (um); Scanning Speed (m/s); Energy Density(J/mm3). An article from October 2021 published in Materials – “A Critical Review on Effect of Process Parameters on Mechanical and Microstructural Properties of Powder-Bed Fusion Additive Manufacturing of SS316L” provides a comprehensive overview of them. (lines 148-154)
- Our manuscript has a completely different concept. As I have already written, the presentation of the effect of the focus level parameter as a parameter describing the relationship between the melting level position and the laser focusing level is clearly unique. As part of the literature review, we also searched through the MDPI database and did not find an article that would mention the focus level parameter or even evaluate its impact on mechanical properties.Our team came across the idea of performing an experiment with the focus level parameter as a variable by accident after a poorly performed service intervention. This will be also incorporated in the article to show difference and novelty. (lines 560-576)
- The scope of the presented results overlaps many individual previous studies with similar topics and also defines itself against these studies.
- We intentionally focused on MDPI database to add references [44-52]
Thank You, Yours sincerely
Zdenek Pitrmuc
